# The Colonization and Effect of *Isaria cateinannulata* on Buckwheat Sprouts

**DOI:** 10.3390/plants12010145

**Published:** 2022-12-28

**Authors:** Xiaona Zhang, Xue Peng, Guimin Yang, Qingfu Chen, Daochao Jin

**Affiliations:** 1The Provincial Key Laboratory for Agricultural Pest Management of Mountainous Region, Institute of Entomology, Guizhou University, Guiyang 550025, China; 2Research Center of Buckwheat Industry Technology, College of Life Science, Guizhou Normal University, Guiyang 550001, China

**Keywords:** *I. cateniannulata*, buckwheat, colonization, condia, defense enzyme

## Abstract

The use of entomogenous fungi as endophytes is currently an area of active research. *Isaria cateniannulata* is an important entomogenous fungus that has been employed for the active control of a range of pests in agricultural and forestry settings, but its direct impact on plants remains to be evaluated. Herein, we assessed the ability of *I. cateniannulata* to colonize buckwheat, *Fagopyrum esculentum* and *F. tataricum*, and its impact on buckwheat defense enzyme activity and physiological indexes. The majority of fungal submerge condia was able to enter into leaves through stomata and veins, and this was followed by conidial attachment, lytic enzyme secretion, conidial deformation, and enhanced defensive enzyme activity within buckwheat, followed by the repair of damaged tissue structures. *I. cateniannulata* populations on buckwheat leaf surfaces (in CFU/g) reached the minimum values at 24 h after inoculation. At this time, the blast analysis revealed that the sequence identity values were 100%, which was consistent with the sequence of *I. cateniannula*. The number of *I. cateniannulata* submerge conidia colonized in the buckwheat leaves gradually rose to peak levels on 7 d post-inoculation, and then gradually declined until 10 d, at which time the buckwheat plant growth index values increased. This study provided novel evidence that *I. cateniannulata* could be leveraged as an endophytic fungus capable of colonizing buckwheat plants and promoting their growth.

## 1. Introduction

Buckwheat *Fagopyrum esculentum* and *F. tataricum* (Polygonaceae) are food sources and they are also used for various medicinal purposes. Buckwheat seeds are rich in high-quality protein, dietary fiber, essential fatty acids, vitamins, minerals, and bioactive sterols and phenolic compounds that have been found to have a valuable function in improving circulation, alleviating gastrointestinal discomfort and preventing the development of tumors. Buckwheat is thus considered to be a functional food [1,2,3,4,5]. Buckwheat sprouts and microgreens have also been recently marketed as a novel food [6,7], containing more phytonutrients, including flavonoids, phenolic compounds, rutin, and quercetin, than buckwheat seeds [8,9,10,11]. Though some effective approaches to promote buckwheat growth and cultivation have been established, more work is needed to optimize sprout yields [7,8]. 

*Isaria cateniannulata* (Liang) Sanmson & Hywel-Jones (Hypocreales: Clavicipitaceae), also known as *Paecilomyces cateniannulatus* or *Cordyceps cateniannulata* [12,13,14], is an important pathogenic fungus of insects that has been utilized for the biocontrol of agricultural and forestry pests [15,16,17,18,19,20,21]. Recently, some scholars have found that it can promote the growth of buckwheat, tomato, tobacco, and so on [22,23,24,25], but the method of to colonizing plants and promoting plant growth has not been explored yet.

Taking buckwheat sprouts as the materials, we explored the colonizing process of *I. cateniannulata* on buckwheat leaves with molecular identification and assessed its impact on resistance enzyme activity and physiological indices in buckwheat plants, in order to understand potential of *I. cateniannulata* as an endophytic fungus.

## 2. Results and Analysis

### 2.1. Colonizing Process of I. cateniannulata on Buckwheat Sprouts via SEM

The ability of *I. cateniannulata* to colonize the leaves of six different buckwheat cultivars was assessed via SEM (Figure 1). The colonization began by 0.5 h after inoculation in all cultivars, and the duration of completing colonization varied among cultivars. The colonization process was completed within 4 h in cultivars No.1412-1 and NO.HT of *F. esculentum*, while this process took 2 h for *F. esculentum* No.1412-16, 8 h for *F. tataricum* No.M18, and 12 h for *F. tataricum* NO.M13 and NO.M55. Overall, the colonization time for *F. esculentum* was shorter than that for *F. tataricum*. In general, the process of *I. cateniannulata* colonization was comparable across buckwheat cultivars. Submerged conidia primarily invaded plant tissues through stomata and veins. During the invasion process, tissues around the submerged conidia were gradually dissolved, potentially by enzymes secreted by conidia [26], or by plant-derived enzymes. Deformed conidia then entered into leaves or were dissolved, leaving dissolution circles that gradually disappeared over time. Variability in the dynamics of this healing process among cultivars may be associated with differences in the ability of particular cultivars to repair colonization-associated wounds. In some cases, a small number of conidia and hyphae attached to leaf surfaces did not colonize the inside of leaves, which might be due to the sites not being suitable for invasion, although more work will be needed to verify this possibility. 

### 2.2. Colonizing Process of Isaria cateniannulata on Buckwheat Sprouts via FM

The colonization of buckwheat sprout leaves by *I. cateniannulata* was assessed via FM (Figure 2). The fluorescence was primarily distributed around veins and stomata throughout the colonization process of the fungus, with fluorescence intensity varying from weak to strong. The fluorescence durations were different among buckwheat cultivars, with a 4 h for No.1412-1 and NO.HT of *F. esculentum*, 2 h for *F. esculentum* No.1412-16, 8 h for *F. tataricum* No.M18, and 12 h for *F. tataricum* NO.M13 and NO.M55, consistent with our SEM findings. After conidia colonized inoculated leaves, the overall brightness of these leaves increased significantly as compared to the control leaves, suggesting that *I. cateniannulata* primarily colonized the inside of buckwheat leaves through veins and stomata. In addition, some bright fluorescent spots were present on leaf surfaces at later time points, consistent with the presence of residual spores in line with our SEM findings. 

### 2.3. Molecular Identification of Isaria cateniannula by Phylogenetic Analysis

The sequence length of the endophytic strain, namely 19GZAl-1.1, obtained from buckwheat sprouts was 563 bp. The ITS data generated for the endophytic isolate was identified with 19GZAl-1 depositing in GenBank with the accession number MW750414. The phylogenetic analysis revealed that the sequence identity values were 94% (Figure 3), identical to sequences of *I. cateniannula*. In the phylogenetic tree, 19GZAl-1 formed an independent branch from the other strains of the *Isaria* genus. Thus, the molecular and phylogenetic analyses of the sequenced data supported the endophytic isolate as *I. cateniannula*, which had colonized in the buckwheat sprouts at 24 h. 

### 2.4. Population Dynamics of I. cateniannulata on Buckwheat Leaf Surfaces

Population dynamics of *I. cateniannulata* on the leaves of different buckwheat cultivars are shown in Figure 4. Fungal levels on leaf surfaces were lowest on the first day of the colonization, in line with SEM and FM findings as a consequence of most conidia having entered into the leaf. These CFU counts then increased over time until peaking on day 7, at which time the CFU/g values for HT, 1412-1, 1412-16, M13, M18, and M55 were 7.2 × 10^4^ CFU/g, 1.1 × 10^5^ CFU/g, 1.9 × 10^5^ CFU/g, 1.3 × 10^5^ CFU/g, 2.0 × 10^5^ CFU/g, and 4.5 × 10^5^ CFU/g, respectively. After day 7, fungal levels declined gradually until day 10, at which time the values for these cultivars were 5.3 × 10^4^ CFU/g, 5.5 × 10^4^ CFU/g, 6.2 × 10^4^ CFU/g, 7.2 × 10^4^ CFU/g, 1.3 × 10^5^ CFU/g, and 1.4 × 10^5^ CFU/g, respectively. This is due to the formation diverse of mutualistic symbiosis between buckwheat leaves and *I. cateniannulata* with the time [27]. Differences in CFU among cultivars may be due to differences in the chemical substances within the leaves of these different plants, although further research will be needed to verify this hypothesis. 

### 2.5. The Impact of I. cateniannulata on Buckwheat Defense Enzyme Activity

Defensive enzyme activity levels of five different enzymes in these six buckwheat cultivars were in the process of changing, either increasing or decreasing, revealing that the effects of *I. cateniannulata* on defense enzyme activities were inconsistent in the tested cultivars. Activity levels of four enzymes, SOD, POD, CAT, and PAL in the treated group were higher than those in the control group, while that of PPO was in contrast. Taking *F. tataricum* No.M13 and *F. esculentum* 1412-16 as an example, the change in enzyme activities in the treated groups and the control groups are shown in Figure 5.

Superoxide dismutase (SOD), peroxidase (POD) and catalase (CAT) are mainly responsible for the production and elimination of intracellular reactive oxygen species (ROS). Under normal circumstances, SOD can dismutate superoxide anion radical to H_2_O_2_, which is further decomposed by enzymes such as CAT and POD [28]. Defense enzyme activity levels of SOD, POD, and CAT in the leaves of *F. tataricum* and *F. tataricum* were higher than those in the control group at 24 h after the colonization of *I. cateniannulata* (Figure 5). The activities of SOD, POD and CAT increased first and then decreased with the time, indicating that *I. cateniannulata* could mediate protective enzyme activities in plants. SOD reached the maximum on 3 d (Figure 5a,f), POD and CAT reached their maximum values on 5 d (Figure 5b,c,g,h), indicating that SOD activity was activated first; POD and CAT played their roles later. However, the activity of CAT decreased, and the activity of POD was stable as time went on, indicating that the concentration of ROS decreased at this time, and POD played a major role.

The activity of the PAL enzyme was on the rise (Figure 5e,j), indicating that the activity of the PAL enzyme could be activated by *I. cateniannulata*. Phenylalanine ammonia-lyase (PAL) can promote the production of salicylic acid, lignin, plant protectants, and other substances through phenylpropane metabolism. The increase of PAL activity could increase the contents of these chemicals in plants, and consequently enhance its resistance to disease [29]. The results indicated that *I. cateniannulata* could increase the buckwheat’s resistance to disease. The activity of the PPO enzyme was lower than that of the control group (Figure 5d,i), which might be due to inhibiting substances of the PPO enzyme in buckwheat having been activated by *I. cateniannulata*. Polyphenol oxidase is the main enzyme that causes browning in many plants. PPO can catalyze the formation of brown quinones from the substrate and participate in the enzymatic browning process of fruits and vegetables. Inhibiting the PPO enzyme has been an essential measure for long storage of fresh fruit and vegetable [30]. The results indicated that *I. cateniannulata* could play an important role in solving the browning problem of buckwheat sprouts during storage.

### 2.6. The Impact of I. cateniannulata on Buckwheat Sprout Growth

The buckwheat growth indices in each treatment group were better than those in the corresponding control group at 10 d after foliar spraying with *I. cateniannulata*, which indicated that the treatment of *I. cateniannulata* was conducive to the growth of buckwheat. However, the differences among different varieties were distinct, and the differences among different cultivars of the same variety were small, indicating that *I. cateniannulata* had different effects on the two buckwheat varieties. Taking *F. tataricum* No.M13 and *F. esculentum* 1412-16 as an example, the change in enzyme activities in the treated groups and the control groups are shown in Figure 5. 

There were no significant differences in the main stem diameter (Figure 6a) of *F. tataricum* No.M13 and in the plant height (Figure 6b) of *F. esculentum* 1412-16 between the treated group and the control group (the detailed data given in Table 1). The main stem diameter values in the treated group were 58.6%, higher than those in the control group of *F. esculentum* (Figure 6a). The average plant heights in the treated group were 28.2%, higher than those in the control group for the M13 of *F. tataricum* (Figure 6b). *I. cateniannulata* could thus positively impact the main stem diameter (Figure 6a), the plant height in *F. tataricum* sprouts, the fresh weight (Figure 6c), the leaf number (Figure 6d), the leaf surface area (Figure 6e), the root length (Figure 6f), the root volume (Figure 6g), and the average root diameter (Figure 6h). These further indicated that *I. cateniannulata* might be beneficial to plant photosynthesis and nutrient transport, thereby enhancing the growth of these buckwheat sprouts.

## 3. Materials and Methods

### 3.1. Isaria cateniannulata Preparation

The *Isaria cateniannulata* strain 19GZAl-1, originally isolated from hymenoptera larvae in Baiyi, Guiyang, Guizhou, in 2019, was provided by the Research Center of Buckwheat Industry Technology, Guizhou Normal University, Guiyang, Guizhou, China. It was originally isolated from hymenoptera larvae in Baiyi, Guiyang, Guizhou, in 2019, and it was maintained on Potato Dextrose Agar medium (PDA). Submerged conidia were produced in a liquid medium containing Potato Dextrose (PDA without agar) and 10 glass beads at 25 ± 1 °C under 16L:8D, followed by incubation for 7 days with shaking at 150 rpm. Submerged conidia were then prepared in suspension, with conidia concentrations being measured via hemocytometer and adjusted to 2 × 10^6^ conidia/mL using sterile water. The viability of the submerged conidia was confirmed to be >90% on PD at >90% relative humidity [15,16,17,18].

### 3.2. Fluorescein Diacetate (FDA) Preparation

FDA (4 mg; Sigma Chemical: St. Louis, MO, USA) was dissolved in 1 mL of acetone. A working solution was freshly prepared for each experiment by adding 35 μL of FDA liquid to 4 mL of deionized water. Samples were maintained on ice, protected from light [15] and were stored at 4 °C for up to 6 months protected from light [25]. 

### 3.3. Buckwheat Sprout Preparation 

Six buckwheat cultivars (three of *F. esculentum*: HT; 1412-1; 1412-16. three of *F. tataricum*: M13; M18; M55) were obtained from the seed bank of the Research Center of Buckwheat Industry Technology, Guizhou Normal University. Buckwheat sprouts with uniform size and shape were disinfected by sequentially treating them with sterile water for 1 min, 75% ethanol for 1 min, 25% NaClO for 1 min, and sterile water for 1 min. Each of three seeds was added to individual holes in a seedling tray (50 holes, each 4.5 cm × 4.5 cm × 4.5 cm) filled with sterilized crystal sand (size 1–2 mm), after which all were treated with an equivalent dose of Dvordo concentrated plant nutrient solution (1:300 nutrient solution: sterile water). Six hundred seeds per cultivar were planted in four trays (3 seeds/holes × 50 holes × 4 trays). The trays were stored in a sterile artificial climate chamber at 25 ± 1 °C with 65–70% relative humidity and a 16 h/8 h light/dark cycle. The nutrient solution was used to water plants every 3 days. After 15 days, sprouts with 2–3 were examined [23].

### 3.4. Treatment of Buckwheat Sprouts with I. cateniannulata 

Buckwheat sprouts of each cultivar in 2.3 were separated into two groups. One was the treatment group and sprayed with 0.5 mL of the prepared submerged *I. cateniannulata* conidia suspension on leaf surfaces, while another was the control group and sprayed with 0.5 mL of sterile water. The other conditions were same to that in Section 3.3.

### 3.5. Observation with Scanning Electron Microscopy (SEM) and Fluorescence Microscopy (FM) 

For the SEM or FM experiments, upper portions of treated and control plants were sampled at 0.5, 1, 2, 4, 8, and 12 h after inoculation, with 5 plants per cultivar being analyzed. Buckwheat sprout leaves were cut to 8 mm × 5 mm and immersed in an alcohol solution after gradient dehydration. The samples were then dried by supercritical drying and put on a tray with double-sided tape, after which they were sputter-coated with carbon (HUS-5GB) using a high vacuum evaporator and then observed via SEM (JSM-6490LV) at an accelerated voltage of 15 kV. Alternatively, leaves were stained with FDA and assessed via an inverted microscope (Nikon, Tokyo, Japan, ECLIPSE Ti-U) at a wavelength of 450–490 nm. Images were taken using a digital camera controller (Nikon Digital Sight PS-U3, Japan).

### 3.6. Analysis of I. cateniannulata on Buckwheat Leaf Surfaces

The colonized number of *I. cateniannulata* liquid spores on buckwheat sprout surfaces were assessed at 3 min, 1 d, 3 d, 5 d, 7 d, 9 d, and 11 d after inoculation. Briefly, 0.1 g samples of leaves were isolated, cut into pieces, added to 1 mL of 0.05% Twin-80, and vortexed for 10 min to achieve a uniform suspension. Conidia were then subjected to gradient dilution, and 200 µL of each gradient suspension was transferred to solid selective medium, with three replicates per gradient concentration. Coated Petri dishes were cultured at 25 ± 1 °C with 70% relative humidity and a 16 h/8 h light/dark cycle. Numbers of colony-forming units (CFUs)/mL were determined as follows: CFUs/mL = average colonized number of three replicates × number of dilutions × 5 times. Data were analyzed using SPSS19.0, and figures were constructed using Origin85.

### 3.7. Molecular Identification of I. cateniannulata in Buckwheat Sprouts

The buckwheat sprouts treated with *I. cateniannulata* for 24 h were sterilized. The disinfection procedures comprised the following steps: washing 1 time with sterile water, immersing in 70% alcohol for 1 min, cleaning with 25% NaClO for 1 min, cleaning the residual NaClO with sterile water for 3 times, and taking 1 drop of the last clean sterile water for culture until no fungi grew for 5 consecutive days; this was considered to be a successful disinfection. The sterilized sprouts were continuously cultured on PDA medium at 25 ± 1 °C for 3 days until they grew out fungi for use.

Total genomic DNA was extracted from 200 mg mycelium of pure isolation obtained after 3 d of growth in buckwheat sprouts treated with *I. cateniannulata*, by using the DNeasy Plant mini kit (Aidlab), following the manufacturer’s instructions. Amplification reactions were performed in volumes of 50 µL, containing template 2 µL, primer1 2 µL, primer4 2 µL, 2 × Master Mix 25 µL, and ddH2O 19 µL. The internal transcribed spacer was amplified using the ITS1/ITS4 primer pair, namely ITS1 (5′-TCCGTAGGTGAACCTGCGG-3′) and ITS4 (5′-TCCTCCGCTTATTGATATGC-3′). The PCR amplification condition was initiated at a 95 °C denature for 6 min, followed by 35 cycles of denaturing at 94 °C for 30 s, annealed at 53 °C for 30 s, and extended at 72 °C for 50 s, then followed by a final extension step at 72 °C for 10 min. Sequencing was performed by Shanghai Sangon Biotech (Shanghai, China) using the above primers. The sequences were stored in GenBank and compared with multiple sequences alignment and manual correction using Clustalw. The phylogenetic tree was constructed using the neighbor-joining method (NJ) in MEGA 7. Confidence values for respective branches were determined by bootstrap analysis (1000 replications) [13]. 

### 3.8. Impact of I. cateniannulata on Buckwheat Defense Enzyme Activity

In total, 30 plants were sampled at 2, 4, 8, 12, and 24 h in both the control and treated groups. The aboveground (stems and leaves) and belowground (root) portions of these plants were separated, washed, and ground to a fine powder in a mortar cooled with liquid nitrogen. Next, 0.1 g samples were added to 1 mL of pre-cooled 50 mmol/L phosphoric acid buffer (containing 1% polyvinylpyrrolidone and 0.2% Ethyl Alcohol, pH 7.8), and were ground on ice until homogenized. Samples were then added to 2 mL tubes and were spun for 10 min at 8000 rpm at 4 °C. Supernatants were then used to measure catalase (CAT), superoxide dismutase (SOD), polyphenol oxidase (PPO), phenylalanine ammonia-lyase (PAL), and peroxidase (POD) activities. All extraction procedures were conducted based on protocols provided in a kit obtained from Suzhou Keming Biotechnology Co., Suzhou, China, LTD. Data were analyzed using SPSS19.0, and figures were constructed using Origin85.

### 3.9. Impact of I. cateniannulata on Buckwheat Growth

The treated and control buckwheat sprouts were cultured for 10 days when sampling. In total, 10 samples were taken from each group, and physical indexes such as plant height, stem diameter, number of leaves, leaf area, fresh weight, root length, root surface area, and root diameter were measured. All treatments including controls had three replicates. The data were measured with a root scanner (GXY-A) and were analyzed using SPSS19.0. 

### 3.10. Statistical Analysis

All statistical analyses were carried out using SPSS v.19.0. Significance between the treatments and controls was evaluated using a t test and One-Way ANOVA at a significance level of 0.05. 

## 4. Discussion

Endophytic fungi are capable of colonizing into host plants through various ways, thereby enhancing their growth [31,32,33]. Endophytes leveraged to date include *Beauveria bassiana* [34,35,36,37,38], *Metarhizium anisopliae* [39,40,41], *Isaria fumosorosea* [42,43], *M. robertsii* [44,45], and *Paecilomyces lilacinus* [46,47]. These fungi have been used to colonize corn [32], the common bean [48], arabidopsis [49], tomatoes [50], wheat, lemen [51], vicia faba [52], lettuce [53], tobacco [22] and other crops [51] via root irrigation [54], seed soaking [32], seed coating [55], stem base injection [56], root or rhizome soaking [46], leaf spraying [56], root mixing, and other methods [57]. In this study, we sprayed the buckwheat sprouts with *I. cateinannulata*, and found it could colonize and promote the growth of buckwheat. This result was consistent with the results of previous studies. Xu et al. found that *I. cateinannulata* could not only increase plant biomass, but also enhance plant stress resistance to abiotic factors [22]; Peng et al. found that it could increase the germination index of seeds of buckwheat [23]. Guan et al. found that it can increase tomato biomass [24]. Our results once again prove that *I. cateinannulata* could be used as an endophytic fungal preparation to colonize plants and promote plant growth.

Entomogenous fungi can colonize plants as endophytes by establishing a mutualistic relationship with host plants without causing any significant disease symptoms [58]. Understanding how these entomogenous fungi colonize plants is thus essential. The previous studies showed that the colonization process and pattern, the precise internal locations and status, of the endophytic fungus could be well evaluated by using light microscopy, electron microscopy and transmission electron microscopy, with the examining of the cellular anatomy of host plants [59,60,61,62]. Herein, we used FM and SEM to assess the colonization process of *I. cateinannulata* on the buckwheat sprout leaf. We determined that conidia invade these leaves via the stomata and veins, dissolving the surrounding tissues and deforming to generate a dissolution circle prior to entering the leaves, after which dissolution circles healed. Terna et al. found that the infection of corn plants by endophytic *P. citrinum* showed mild histopathological responses in host tissues [59]. Our findings offered more detail insight into the colonization process. In addition to direct observation of fungal colonization, molecular detection is another effective means to verify that endophytic fungi colonize plants [63,64]. Our sequenced data verified the endophytic isolate as *I. cateniannula*, colonized in the buckwheat sprouts at 24 h. 

One of the most striking features of entomogenous fungi is their ability to colonize diverse hosts, ranging from obligate pathogens to very broad facultative pathogens [65]. The ability of these fungi to adapt to diverse hosts may be associated with their ability to obtain nutrients from sources other than insects, enabling them to colonize plants [66,67]. In this study, we found that the *I. cateinannulata* colonization process was similar in two buckwheat species, indicating that this endophyte was able to absorb nutrients under a range of growth conditions. Moreover, colonization times were shorter in *F. esculentum* than in *F. tataricum*, indicating that *F. esculentum* is more suitable than *F. tataricum* for *I. cateinannulata* colonizing. 

The duration of colonization and the levels of entomogenous fungi within plant leaves are key factors that determine the efficacy of these fungi as mediators of pest control and plant growth. Cui et al. found that higher concentrations and longer residue times of *B. bassiana* on maize leaves could result in significant impacts on physical indexes such as plant height, leaf length, and leaf width of maize [55]. Zhang et al. found that *I. cateniannulat* attached to the surfaces of vegetable leaves could continuously provide protection against *Tetranychus urticae* for over 10 days [15]. Wakil et al. found that the integrated application of endophytic colonized *B. bassiana* and *B. thuringiensis* might be an effective approach against *H. armigera* [68]; Qayyum et al. found that the endophytic colonization of *B. bassiana* had an effective strategy to control *H. armigera* in tomatoes [69]. Our results were consistent with these previous studies. We found that *I. cateniannulata* could stay on the surface of buckwheat leaves for over 10 days, with relatively high residual concentrations that significantly enhanced plant growth with higher physical indexes such as plant height, stem diameter, leaf area, root length, and fresh weight. However, we observed cultivar-specific differences in the duration and levels of *I. cateniannulata* presence on leaf surfaces in this study, which might be due to the submerged conidia having been affected by the chemical composition on the leaf surfaces of the tested cultivars or by composition with other endophytic fungi inside the buckwheat, leading to the different colonization rates of *I. cateniannulata* in buckwheat [70].

Colonization of entomogenous fungi on/in plants can not only prevent pathogen-induced damage, but also protect plants from consumption by herbivores. The mechanistic basis for such protection is complex and is at least partially associated with the enhancement of the defense enzyme activity in host plants. For example, *B. beauveria* colonization in maize seedlings was associated with the enhancement of CAT, POD, SOD, and PAL activity [55], thereby protecting maize from corn borer-induced damage [32]. The increase of PAL activity can mediate the increase of the contents of salicylic acid, lignin and plant protectants and accordingly enhance the disease resistance of the plant [29]. Some studies have shown that PAL could enhance the resistance of the soybean against the nematode *Heterodera glycines* [71]. Our findings are in line with these documentations, as we found that *I. cateniannulata* colonization enhanced the activity of these defense enzymes in a cultivar-specific manner. The variation in endophyte-induced defensive responses among these cultivars may be related to differences in antioxidant contents in these different cultivars [21]. Buckwheat exhibits a robust antioxidant capacity, particularly under adverse conditions, protecting cells by SOD, CAT, and POD rapidly removing the free radicals and associated intermediates [72]. Due to PPO participates in the enzyme-mediated browning process of fruits and vegetables, inhibiting PPO activity has been a major measure to solve the browning problem of stored fresh fruit and vegetables [30]. Our results were consistent with the that of previous studies and indicated that *I. cateniannulata* could be used as a storage preparation to extend shelf life of buckwheat sprouts.

## 5. Conclusions

These results are the novel evidences that *I. cateniannulata* can colonize buckwheat sprouts. Fungal spores were able to colonize buckwheat plants within 12 h and exhibited a leaf residual time of over 10 days. Importantly, the endophytic *I. cateniannulata* could promote buckwheat growth and defense enzyme activity, although the mechanistic basis for this activity remains to be explored. Overall, our data offer a novel insight into the relationship between entomogenous fungi and plants, providing a foundation for future studies of *I. cateniannulata* as a biological agent.

## Figures and Tables

**Figure 1 plants-12-00145-f001:**
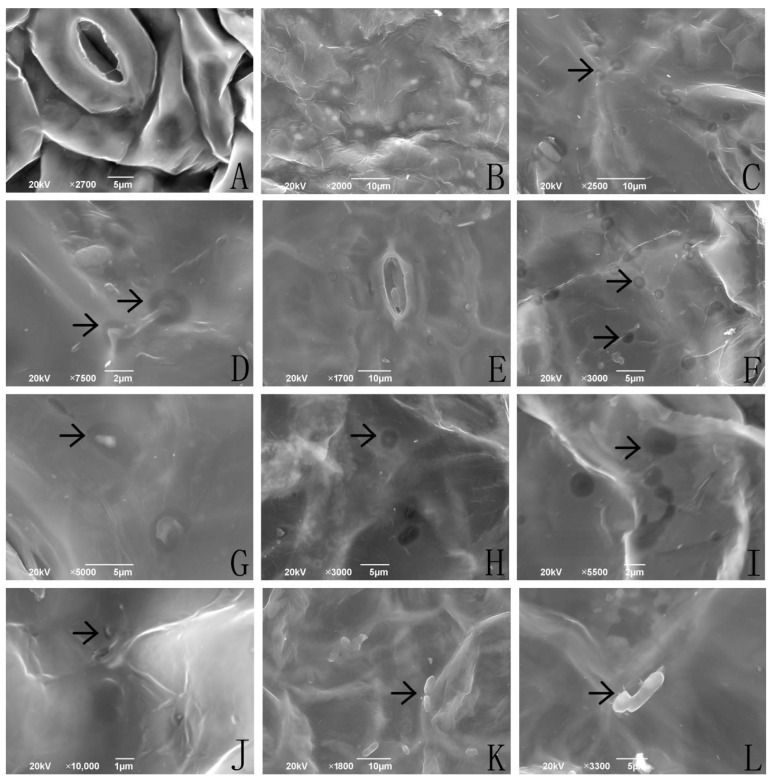
*I. cateinannulata* submerged conidia colonization of buckwheat leaf as assessed via scanning electron microscopy. Note: (**A**) Control group, no attachment or submerged conidial colonization were observed during the study period. (**B**) Submerged conidia (diameter ≤ 2 µm) were attached to leaf surfaces after spraying. (**C**) Submerged conidia on leaf surfaces, at 0.5 h after inoculation, began to invade leaf tissues, with clear evidence of tissue dissolution surrounding spores, which were primarily at leaf veins and stomata. (**D**) A germ tube was oriented in place and had transferred its nucleus into the leaf. (**E**,**F**) At 1–2 h after inoculation, the dissolution area of leaf tissue surrounding submerged conidia increased significantly, and these conidia gradually deformed and shrunk. (**G**–**J**) At 2–4 h after inoculation, the dissolution area of tissue surrounding inoculated spores gradually decreased, and numbers of submerged conidia had fallen substantially. The majority of conidia had either entered into leaves or dissolved with pit-shaped dissolution circles at this time point. Leaf morphology was normal at 4–8 h after spraying, with no morphological differences between inoculated leaves and control leaves. (**K**,**L**) Leaf tissues after inoculation. Some conidia (diameter > 2 µm) did not enter the leaves and formed no visible dissolution circles, instead growing to yield tentacle-like structures affixed to the blade surface (1 L). arrows: the condition of the *I. cateinannulata* submerged conidia and the buckwheat leaf.

**Figure 2 plants-12-00145-f002:**
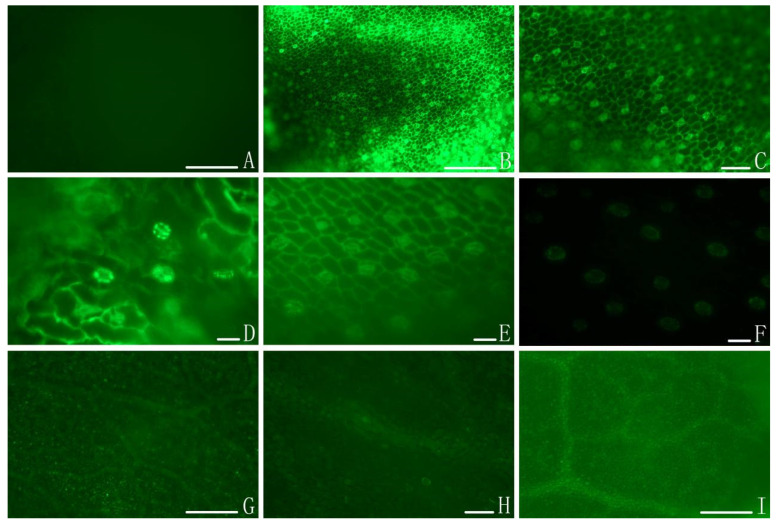
The colonization of *I. cateinannulata* submerged conidia in buckwheat leaf as assessed via fluorescence microscopy. Note: (**A**) In control samples, no fluorescent aggregation was observed, consistent with an absence of submerged conidia colonization. (**B**–**D**) At 0.5 to 1 h after spraying, significant fluorescent signal was visible and distributed around veins and stomata. (**E**,**F**) At 2 to 8 h after spraying, the fluorescence intensity of veins and stomata decreased. (**G**–**I**) At 4 to 12 h after spraying, fluorescence was no longer visibly associated with veins or stomata, and the leaves presented a uniform color brighter than that of control leaves (**I**). Some bright spots were visible on the blade surface (**G**,**H**), which may correspond to small amounts of conidia attached to the blade surface. Scale bar: 1 mm.

**Figure 3 plants-12-00145-f003:**
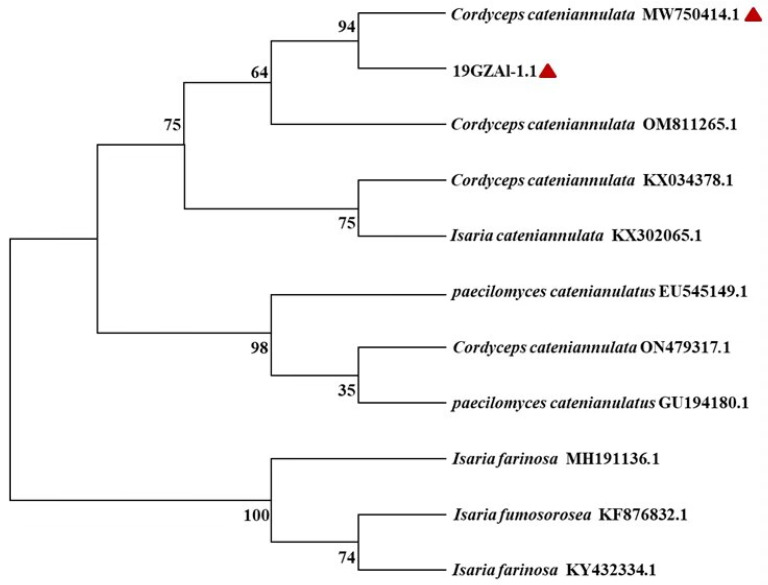
Phylogenetic analysis of the partial rDNA-ITS sequences of the endophytic isolated strain and related species (obtained from GenBank). Note: the two marked with red triangles were originally the same strain. 19GZAl-1.1 was the endophytic strain in this study; MW750414: the accession number of 19GZAl-1 *Isaria cateniannula* in GenBank submitted by the authors.

**Figure 4 plants-12-00145-f004:**
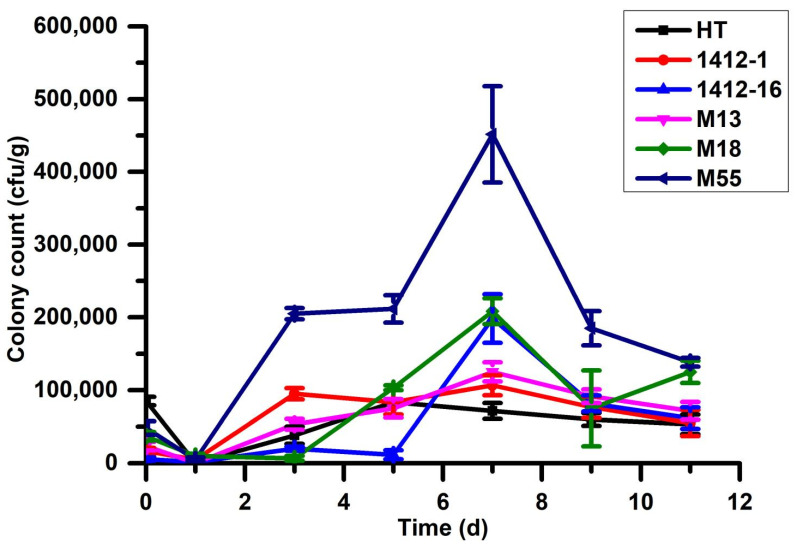
Population dynamics of *I. cateinannulata* on the leaf surfaces of six buckwheat cultivars (in CFUs/g).

**Figure 5 plants-12-00145-f005:**
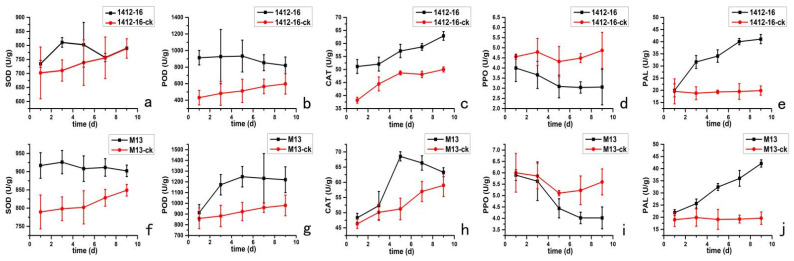
The impact of *I. cateinannulata* on defense enzyme activity in the leaves of two buckwheat varieties, *F. tataricum* No.M13 and 1412-16. Note: (**a**,**f**) Changes in SOD activity; (**b**,**g**): Changes in POD activity; (**c**,**h**) Changes in CAT activity; (**d**,**i**) Changes in PAL activity; (**e**,**j**) Changes in PPO activity; ck: the control group.

**Figure 6 plants-12-00145-f006:**
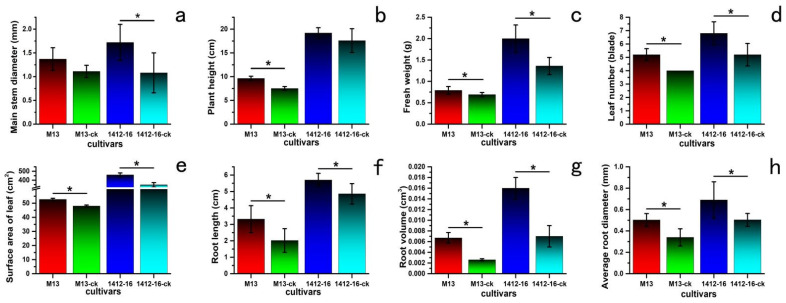
The impact of *I. cateniannulata* on the growth of two buckwheat varieties, *F. tataricum* No.M13 and *F. esculentum* 1412-16. at 10 d after foliar spraying. Note: significant differences between treatment group and control group were compared by one-way ANOVA. Significant differences at *p* < 0.05 were marked as single (*). The data were calculated using ten replicate assays. (**a**) main stem diameter; (**b**) plant height; (**c**) fresh weight; (**d**) leaf number; (**e**) surface area of leaf; (**f**) root length; (**g**) root volume; (**h**) average root diameter. ck: control group.

**Table 1 plants-12-00145-t001:** The impact of *I. cateniannulata* on the growth of different buckwheat cultivars at 10 d after foliar spraying.

Cultivars	Plant Height (cm)	Main Stem Diameter (mm)	Fresh Weight (g)	Leaf Number	Surface Area of Leaf (cm^2^)	Root Length (cm)	Root Surface Area (cm)	Root Volume (cm^3^)	Average Root Diameter (mm)
M13-s	9.6400 ± 0.4561 *	1.3700 ± 0.2421	0.7960 ± 0.0839 *	5.20 ± 0.45 *	52.7446 ± 0.7642 *	3.3200 ± 0.8228 *	0.6271 ± 0.2157 *	0.0067 ± 0.0011 *	0.5025 ± 0.0612 *
M13-d	7.5200 ± 0.4087	1.1140 ± 0.1324	0.6900 ± 0.0579	4.00 ± 0.00	48.0357 ± 0.7185	2.020 ± 0.7259	0.2464 ± 0.0171	0.0026 ± 0.0002	0.3400 ± 0.0876
M18-s	13.2000 ± 0.5745 *	1.7500 ± 0.2749	1.5480 ± 0.0817 *	5.60 ± 0.89 *	98.0285 ± 7.0741 *	4.6000 ± 0.2449 *	0.7382 ± 0.0989 *	0.0043 ± 0.0001 *	0.4848 ± 0.0728 *
M18-d	11.5600 ± 1.0945	1.4140 ± 0.3620	1.3540 ± 0.0288	3.60 ± 0.54	74.5337 ± 9.8254	4.3400 ± 0.0894	0.4018 ± 0.2003	0.0026 ± 0.0012	0.3730 ± 0.0325
M55-s	10.5800 ± 1.0426 *	1.5280 ± 0.2774	1.4160 ± 0.0709 *	4.60 ± 0.89 *	84.0771 ± 5.8239 *	2.5200 ± 0.3898 *	0.6229 ± 0.1683 *	0.0066 ± 0.0025 *	0.5662 ± 0.08581 *
M55-d	9.6600 ± 0.3782	1.2800 ± 0.1756	1.2840 ± 0.0709	3.80 ± 0.45	54.5186 ± 5.1692	2.1200 ± 0.2683	0.2198 ± 0.0531	0.0026 ± 0.0011	0.3238 ± 0.0347
HT-s	30.1400 ± 4.3478	3.2040 ± 0.3325 *	3.6140 ± 1.0313 *	7.00 ± 0.71 *	187.3840 ± 14.5548 *	8.7800 ± 1.4772 *	1.1236 ± 0.6708 *	0.0163 ± 0.0022 *	0.4456 ± 0.0884 *
HT-d	27.2200 ± 3.4737	2.1640 ± 0.2154	1.9160 ± 0.4852	4.80 ± 0.84	155.1280 ± 20.4564	4.3600 ± 0.1816	0.8348 ± 0.0847	0.0040 ± 0.0004	0.3812 ± 0.0522
1412-1-s	15.8400 ± 2.4470	1.7700 ± 0.0906 *	1.7560 ± 0.9965 *	6.00 ± 1.22 *	212.8579 ± 31.3140 *	4.1800 ± 0.1923 *	0.6979 ± 0.0583 *	0.0131 ± 0.0037 *	0.5895 ± 0.0604 *
1412-1-d	11.0400 ± 1.0479	1.5320 ± 0.1223	1.4300 ± 0.1288	4.80 ± 0.84	164.9717 ± 9.7931	3.3200 ± 0.1788	0.4290 ± 0.0581	0.0050 ± 0.0028	0.4583 ± 0.0275
1412-16-s	19.2000 ± 1.0901	1.7200 ± 0.3843 *	2.0000 ± 0.3221 *	6.80 ± 0.86 *	459.2300 ± 22.0052 *	5.700 ± 0.4000 *	0.7692 ± 0.0602 *	0.0161 ± 0.0026 *	0.6941 ± 0.1762 *
1412-16-d	17.6000 ± 2.4749	1.0840 ± 0.4287	1.3660 ± 0.2017	5.20 ± 0.84	347.5329 ± 25.0809	4.8600 ± 0.6107	0.4587 ± 0.2643	0.0070 ± 0.0023	0.5041 ± 0.0649

Note: Significant differences between the treatment group and the control group were compared by one-way ANOVA. Significant differences at *p* < 0.05 were marked as single (*). The data (mean ± SD) were calculated using ten replicate assays.

## Data Availability

The data that support the findings of this study are available from the corresponding author and the first author upon reasonable request.

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
