# Peer review of "The Colonization and Effect of Isaria cateinannulata on Buckwheat Sprouts"

_plants, 2022, doi:10.3390/plants12010145_

Round 1
Reviewer 1 Report
This ms is within the scope of journal, the authors assessed the colonization of endophytic Isaria cateniannulata in buckwheat, Fagopyrum esculentum and F. tataricum, and impact on buckwheat defense enzyme activity. Based on the results the authors tried to prove that Isaria cateniannulata could be colonized in buckwheat.
Overall the write-up of ms is weak and it is advised the authors should seek help from English native colleagues for the fluency in language. The objectives of the study are clear, the experimental design is appropriate and the results support the conclusion. The authors generated good data but I have some concerns in ms like the authors should extend the Introduction section providing the colonization ability of “Isaria cateniannulata” and impact on the growth of other plants etc. It is good to elaborate References 22-24 either in Introduction or in Discussion section. Most of the sub-sections/protocols in Methodology section are not supported with relevant citation/s – it is important to give relevant citations to all the protocols. Further, in assay (sub-section 2.9) the conclusions are based on single, un-repeated experiments and this is not acceptable. Replicates within an experiment measure variability within the system, whereas repetition of the experiments ensures repeatability of the results and lack of artefacts. Would you have the same conclusions if the study was repeated is the question that must be addressed?
I am of the opinion that in the current form this ms can not be accepted for its publication in Plants but the Editor may allow authors for major revisions to improve the article.
Some other suggestions are as follows:
P1, L2; replace “buck-wheat” with “buckwheat”
P1, L43; give complete taxonomy of “Isaria cateniannulata”
P2, L54; was strain 19GZAl-1 identified on molecular basis, give details and how the authors was confirmed that the fungi oozed out from the plant parts was the same – what type of selective media was used to grow fungi without contamination?
P2, L57; give company/brand of Potato Dextrose Agar medium?
P2, L62, what is “PD”?
P2, L65; where is “Sigma Chemical”
P2, L69-82; give relevant citation/s for sprout preparation?
P2, L84-87; give more details for treating the sprouts etc.?
P3, L94 and 96; provide location of Nikon company?
P3, L97; delete “population dynamics”
P3, L98; both word “population” and “dynamics” are not appropriate for fungal colonization in leaves
P3, L143; what is “Origin85”
P4, L145-148; more details about the procedure, treatments, replications and control etc. should be provided?
P4, 149; sub-heading “Statistical analysis” should be included and all factors and statistical package etc. should be given?
P4, L150; delete “and analysis”
P9, L287; give statistics and significance letters?
P10, L292; should be “Discussion” and “Conclusion should be under separate heading at the end?
P10, L297-300; I noticed two relevant citations and authors may consider to include; https://www.frontiersin.org/articles/10.3389/fmicb.2020.522368/full and https://www.sciencedirect.com/science/article/pii/S104996441500050X
Author Response
Response to the Referee 1 1.Overall the write-up of ms is weak and it is advised the authors should seek help from English native colleagues for the fluency in language. Answer: Thank you very much for your valuable advice. The colleagues who are good at English helped us to revise the text that are marked in colour fonts. 2.The authors generated good data but I have some concerns in ms like the authors should extend the Introduction section providing the colonization ability of “Isaria cateniannulata” and impact on the growth of other plants etc. It is good to elaborate References 22-24 either in Introduction or in Discussion section. Answer: Thank you for suggestion. We have added some text in the discussion section of the revised ms. 3.Further, in assay (sub-section 2.9) the conclusions are based on single, un-repeated experiments and this is not acceptable. Replicates within an experiment measure variability within the system, whereas repetition of the experiments ensures repeatability of the results and lack of artefacts. Would you have the same conclusions if the study was repeated is the question that must be addressed? Answer: Thank you very much for your reminding. We did 3 replicants for each tretment. We have added the sentence “All treatments including contols had three replicants” in sub-section 2.9 of the new ms... 4.P1, L2; replace “buck-wheat” with “buckwheat” Answer: Here we have revised it. 5.P1, L43; give complete taxonomy of “Isaria cateniannulata” Answer: We have modified it as “Isaria cateniannulata (Liang) Sanmson &Hywel-Jones (Hypocreales: Clavicipitaceae), also known as Paecilomyces cateniannulatus or Cordyceps cateniannulata [12-14], ......”. 6.P2, L54; was strain 19GZAl-1 identified on molecular basis, give details and how the authors was confirmed that the fungi oozed out from the plant parts was the same – what type of selective media was used to grow fungi without contamination? Answer: In this paper, the spores isolated from plants were identified by both morphological and molecular techniques. The used isolation medium were the medium with double antibodies, and the finally used strain 19GZAl-1 was obtained after multiple isolation from a single strains. Moreover, we have been preparing a paper on selecting and identifying of entomogenous fungal strains. 7.P2, L57; give company/brand of Potato Dextrose Agar medium? Answer: PDA medium was made by ourselves. The method is as follows: 200g potatoes were cut up and boil with water until they become soft, add 20g glucose and 18g AGAR into the liquid after filtering the filter residue, water up to 1000mL, and divide as required. 8.P2, L62, what is “PD”? Answer: Potato Dextrose was PDA medium without agar. We have explained it in the new ms. 9.P2, L65; where is “Sigma Chemical” Answer: The company is in the United States, we have added it. 10. P2, L69-82; give relevant citation/s for sprout preparation? Answer: Thank you for suggestion, but we don’t think it is necessary to give the citation(s) because the preparation methed of buckwheat sprout were presented detaily in the ms. 11. P2, L84-87; give more details for treating the sprouts etc.? Answer: Here we have revised the text. 12. P3, L94 and 96; provide location of Nikon company? Answer: Here we have added the location information. 13. P3, L97; delete “population dynamics” Answer: It has been deleted. 14. P3, L98; both word “population” and “dynamics” are not appropriate for fungal colonization in leaves Answer: We have replaced “The dynamics of I. cateniannulata populations” with “The colonized number of I. cateniannulata liquid spores”. 15. P3, L143; what is “Origin85” Answer: Origin is a graphic visualization and data analysis software developed by OriginLab (formerly Microcal), an advanced data analysis software commonly used by researchers and engineers 16. P4, L145-148; more details about the procedure, treatments, replications and control etc. should be provided? Answer: We did 3 replicants for each tretment. We have added a sentence “All treatments including contols had three replicants” in sub-section 2.9 of the new ms.. 17. P4, 149; sub-heading “Statistical analysis” should be included and all factors and statistical package etc. should be given? Answer: We have modified the text of sub-section “2.10 Statistical analysis” as follows: All statistical analyses were carried out using SPSS v.19.0. Signifcance between the treatments and controls was evaluated using a t test and One-Way ANOVA at a signifcance level of 0.05. 18. P4, L150; delete “and analysis” Answer: It has been deleted. 19. P9, L287; give statistics and significance letters? Answer: We have revised. 20. P10, L292; should be “Discussion” and “Conclusion should be under separate heading at the end? Answer: We have replaced “Conclusion and discussion” with “Discussion”, and added “5. Conclusion ” . 21. P10, L297-300; I noticed two relevant citations and authors may consider to include; https://www.frontiersin.org/articles/10.3389/fmicb.2020.522368/full and https://www.sciencedirect.com/science/article/pii/S104996441500050X Answer: We have carefully read the two recommended papers and cited them in the discussion section.
Reviewer 2 Report
Dear authors, thanks for your manuscript. Your work is exciting and is focused on the use of entomogenous fungi. I recommend major revision since you should rewrite some parts.
When you describe buckwheat, you should insert the terms "common" and "Tartary." I found typos (for example, in the abstract, line 24).
About methods, you should add the statistical method. About SEM, you should add the work condition and some important information about sputtering (critical point drier). If you didn't use critical point or esem conditions, your micrographs could contain artifacts.
Rewrite results and discussion. Then add conclusions
Author Response
Thank you for your comments concerning our manuscript entitled “The colonization and effect of Isaria cateinannulata on buckwheat sprouts”(ID: plants-2041882). The comments are all valuable and very helpful for revising and improving our paper. We have studied comments carefully and have made correction which we hope meet with approval. The revised text are marked in color in the paper.
Reviewer 3 Report
The authors in their manuscript entitled “The colonization and effect of Isaria cateinannulata on buck-2 wheat sprouts”, describe the colonization process of the fungus in different buckwheat cultivars, and different aspects regarding the impact in plant defense and growth using different techniques spanning from molecular and biochemical assays, microscopy observations of cells to macro physical indices.
The approach is standard, the results are descriptive and give a step further knowledge regarding plant’s responses to fungus’ colonization. My comments are not on the essence of the results but on their presentation.
1) In line 106 the authors should describe clearly what numbers “5” and “10” in the equation type, refer to.
2) In paragraph 3.3 (lines 208-214), the authors should describe clearly what 19GZAI1-1 19GZAI1 strains refer to (or originate/occur from). A detailed description is needed.
3) In figure 3, more (closely related) species should be incorporated to expand the phylogenetic tree, also including outgroups (not closely related species).
4) In the legend of figure 3 the fungal IDs (Genebank) should be connected to the relevant species. Putatively this should be done also in image 3 as well.
5) Paragraph 3.5 and figure 5 should be improved (rewritten and better figure presentation) taking in consideration the following:
a. It is a pity the authors do not measure ROS (e.g., H2O2) with a relevant assay to couple the results with the enzymatic activity. Since there are no evidence regarding ROS concentrations are presented, the authors should adapt their description in lines 250-256, in a way declaring that these enzymes “could” do this or that, or the enzyme activities “could” mean this or that, since there are no data from ROS.
b. In figure 5, images and indices should be larger in size in order to be clearly readable from paper print (even in the PC screen is difficult to discriminate the error bars and read the index descriptions in top right corners. It would be also better to present the two cultivars separately (thus the double number of images), probably side by side in order for the differences and error bars to be clearly interpreted. There is definitely a cultivar dependent response for each enzyme, and it would be better to show it clearly and comment on it (in the text) further as well.
c. Paragraph in lines 257-260, should be rewritten and expanded, showing the relevance of PAL to Salicylic acid and plant defense responses/primming. This is the clearest biochemical effect/results this experiment shows but it is underestimated/less presented in the text! Relevant bibliography should be added as well.
d. PPO it is not clear what its role is. Thus, unless the authors connect this enzyme role and the specific effect it has with the relevant argumentation, it is probably better to withdraw this assay-result and image from the manuscript, since it is not clearly supported or connected to the rest of the experiment.
e. An expanded figure legend with a bit more of description of what figure 5 says, would possibly add to the text/presentation
6) Table 1 is difficult to interpret. These data (Table1) should go as supplementary material and a new figure with images regarding each parameter (all 9) should be made instead. This would make easier to see any effect (significant or not) between the different cultivars. The authors should also decide if they prefer to use either the term “cultivar” (see materials and methods) or the term “variety” through the whole of the manuscript.
7) The discussion part in the last two paragraphs (lines 346-365) should be adapted regarding the previous comments (see comments No 5 and 6), particularly regarding the connection of PAL-SA and defense (could be priming of the plants as well), and a bit furthermore regarding the putative cultivar(s) effect.
8) Minor syntax-expression-typographic corrections should be also checked.
Author Response
Response to the Referee 3 1.In line 106 the authors should describe clearly what numbers “5” and “10” in the equation type, refer to. Answer: We have modified the equation. 2.In paragraph 3.3 (lines 208-214), the authors should describe clearly what 19GZAI1-1 19GZAI1 strains refer to (or originate/occur from). A detailed description is needed. Answer: The fungal strain 19GZAI1-1 (no strains numbered 19GZAI1) was originally isolated from hymenoptera larvae in Baiyi, Guiyang, Guizhou, in 2019. We have added this in sub-section “ 2.1. Isaria cateniannulata preparation”. Moreover, We have revised the text about the phylogentic relationships. 3.In figure 3, more (closely related) species should be incorporated to expand the phylogenetic tree, also including outgroups (not closely related species). Answer: We have revised the text about the phylogentic relationships in the sub-section “3.3”. 4.In the legend of figure 3 the fungal IDs (Genebank) should be connected to the relevant species. Putatively this should be done also in image 3 as well. Answer: We have added the fungal species with the GeneBank IDs in the figure 3. 5.Paragraph 3.5 and figure 5 should be improved (rewritten and better figure presentation) taking in consideration the following: a. It is a pity the authors do not measure ROS (e.g., H2O2) with a relevant assay to couple the results with the enzymatic activity. Since there are no evidence regarding ROS concentrations are presented, the authors should adapt their description in lines 250-256, in a way declaring that these enzymes “could” do this or that, or the enzyme activities “could” mean this or that, since there are no data from ROS. Answer: Thank you for helpful suggestion. We have revised the related text. b.In figure 5, images and indices should be larger in size in order to be clearly readable from paper print (even in the PC screen is difficult to discriminate the error bars and read the index descriptions in top right corners. It would be also better to present the two cultivars separately (thus the double number of images), probably side by side in order for the differences and error bars to be clearly interpreted. There is definitely a cultivar dependent response for each enzyme, and it would be better to show it clearly and comment on it (in the text) further as well. Answer: Fig.5 has been redone as you suggested, the two cultivars separateed. The relevant content has been revised yet. c.Paragraph in lines 257-260, should be rewritten and expanded, showing the relevance of PAL to Salicylic acid and plant defense responses/primming. This is the clearest biochemical effect/results this experiment shows but it is underestimated/less presented in the text! Relevant bibliography should be added as well. Answer: Thanks for helpful suggestion. We have rewritten this paragraph. d.PPO it is not clear what its role is. Thus, unless the authors connect this enzyme role and the specific effect it has with the relevant argumentation, it is probably better to withdraw this assay-result and image from the manuscript, since it is not clearly supported or connected to the rest of the experiment. Answer: We have rewritten this paragraph with explaination of PPO function.. e.An expanded figure legend with a bit more of description of what figure 5 says, would possibly add to the text/presentation Answer: In redrew figure 5, with rivised text, the two cultivars’ responses to each enzyme have been showed clearly. It seems no need to expand the figure legend. 6.Table 1 is difficult to interpret. These data (Table1) should go as supplementary material and a new figure with images regarding each parameter (all 9) should be made instead. This would make easier to see any effect (significant or not) between the different cultivars. The authors should also decide if they prefer to use either the term “cultivar” (see materials and methods) or the term “variety” through the whole of the manuscript. Answer: Thank you very much for your valuable advice. According to your suggestions, we have changed “Table 1” into the images, please see Figure 6 in revised ms. “Table 1” is now provided as “Table S1”. 7.The discussion part in the last two paragraphs (lines 346-365) should be adapted regarding the previous comments (see comments No 5 and 6), particularly regarding the connection of PAL-SA and defense (could be priming of the plants as well), and a bit furthermore regarding the putative cultivar(s) effect. Answer: Thanks, we have rivised the related paragraphs, as answer above 8.Minor syntax-expression-typographic corrections should be also checked. Answer: Thank you very much. We checed entire ms carefully.
Round 2
Reviewer 1 Report
I have gone through the revised version of ms and am convinced the authors substantially improve the quality of the article but still need minor language editing and spell check – this is a unique study and will be good addition in the literature on endophytic fungi – I endorse the acceptance of this article for its publication in Plants
Author Response
I have gone through the revised version of ms and am convinced the authors substantially improve the quality of the article but still need minor language editing and spell check – this is a unique study and will be good addition in the literature on endophytic fungi – I endorse the acceptance of this article for its publication in Plants. Answer:Thank you for your approval of the article. We accepted the proposed language editing and spell-checking and have revised the text that are marked in color in the ms.
Reviewer 2 Report
dear authors, tjanks for the revised version. I think that you should add more information about scanning electron microscopy techniques. you should add the voltage and if you observed the samples under low vacuum or high vacuum. if you didn't use the a critical point drier, your anatomical observations could be artifacts.
check typos, for example line 152
best regards
Author Response
1.dear authors, thanks for the revised version. I think that you should add more information about scanning electron microscopy techniques. You should add the voltage and if you observed the samples under low vacuum or high vacuum. if you didn't use the a critical point drier, your anatomical observations could be artifacts. Answer: Thank you very much for your valuable advice. According to your suggestions, we have revised the related text. 2.check typos, for example line 152 Answer: Thank you very much. We had corrected and checked entire ms carefully.
